# Facile Synthesis to Porous TiO_2_ Nanostructures at Low Temperature for Efficient Visible-Light Degradation of Tetracycline

**DOI:** 10.3390/nano14110943

**Published:** 2024-05-27

**Authors:** Peng Lian, Aimiao Qin, Zhisen Liu, Hao Ma, Lei Liao, Kaiyou Zhang, Ning Li

**Affiliations:** 1Key Laboratory of New Processing Technology for Nonferrous Metals & Materials, Ministry of Education, Guangxi Key Laboratory of Environmental Pollution Control Theory and Technology, College of Materials Science & Engineering, Guilin University of Technology, Guilin 541004, China; lianpeng@gdupt.edu.cn (P.L.); fangqiu2001@163.com (L.L.); kaiyou2014@glut.edu.cn (K.Z.); 2College of Chemistry, Guangdong University of Petrochemical Technology, Maoming 525000, China; lzs_415@163.com (Z.L.); thma@gdupt.edu.cn (H.M.)

**Keywords:** nanoporous, low temperature, in situ XRD, density functional theory, degradation routes

## Abstract

In this study, nanoporous TiO_2_ with hierarchical micro/nanostructures was synthesized on a large scale by a facile one-step solvothermal method at a low temperature. A series of characterizations was performed and carried out on the as-prepared photocatalysts, which were applied to the degradation of the antibiotic tetracycline (TC). The results demonstrated that nanoporous TiO_2_ obtained at a solvothermal temperature of 100 °C had a spherical morphology with high crystallinity and a relatively large specific surface area, composed of a large number of nanospheres. The nanoporous TiO_2_ with hierarchical micro/nanostructures exhibited excellent photocatalytic degradation activity for TC under simulated sunlight. The degradation rate was close to 100% after 30 min of UV light irradiation, and reached 79% only after 60 min of visible light irradiation, which was much better than the photodegradation performance of commercial TiO_2_ (only 29%). Moreover, the possible intermediates formed during the photocatalytic degradation of TC were explored by the density functional theory calculations and HPLC-MS spectra. Furthermore, two possible degradation routes were proposed, which provided experimental and theoretical support for the photocatalytic degradation of TC. In this study, we provide a new approach for the hierarchical micro/nanostructure of nanoporous TiO_2_, which can be applied in industrial manufacturing fields.

## 1. Introduction

Nanoporous TiO_2_ spheres have emerged recently as a new class of TiO_2_ nanomaterials for photochemical applications. Compared with conventional TiO_2_ nanoparticles, nanoporous TiO_2_ presents significant advantages in terms of structural isotropy, structural diversity on nano- and (sub)microscales, structural stability, and superior photocatalytic performances; thus, nanoporous TiO_2_ has received much attention from many researchers [1,2,3,4,5,6]. Many synthetic methods, such as sol–gel, hydrothermal coprecipitation, polyol synthesis, etc., have been reported for the synthesis of TiO_2_ nanoparticles. Amongst the reported methods, the hydrothermal synthesis method is currently recognized as the most important chemical method due to its multiple advantages, including the fact that its stability, high homogeneity, and flexibility of processing allow varying reaction parameters [7,8,9]. In recent decades, a great deal of research has been performed to develop nanoporous TiO_2_-based efficient visible light active photocatalysts. These efforts include, amongst others, morphology design, doping, or combining with metal/other semiconductors to form semiconductor heterojunction [10,11,12]. However, there are relatively few reports on the solvothermal synthesis of nanoporous TiO_2_ at a low temperature; the controllable preparation of nanoporous TiO_2_ with hierarchical micro/nanostructures on a large scale with ideal morphology at a low temperature is still considered a great challenge in materials science [13]. Liu et al. reported that nanoporous TiO_2_ was synthesized by titanium tetraisopropanolate (TTIP) and anhydrous acetone at the solvothermal reaction temperature of 200 °C; the as-synthesized spheres (diameter: 0.7–1.3 μm) are comprised of numerous nanoparticles (size: 9–10 nm), which demonstrated excellent photocatalytic performance for both photocatalytic production of hydrogen from water and degradation of methyl orange [14]. Tellam et al. used titanium carbide (TiC) as a raw material to synthesize rutile TiO_2_ at a low temperature (100 °C) by a hydrothermal method. They investigated the effects of synthesis time and temperature on the catalyst properties of synthetic materials. However, it was also confirmed that the degradation rate of catalysts to rhodamine G was relatively low (60%) after being irradiated with visible light for 200 min [15]. Therefore, there is an urgent need to find a synthesis method of nanoporous TiO_2_ with mild reaction conditions, low production cost, and excellent performance, which has important significance and prospects [16,17,18].

With the widespread use of antibiotics in the past few decades, toxic and harmful persistent organic pollutants [19,20], such as tetracycline (TC), in wastewater exert increasingly serious impacts on the environment. TC wastewater is characterized by complex composition, large water quality fluctuations, and strong biological toxicity. Further, the TC deluge contributes to the emergence of antibiotic-resistant bacteria and antibiotic-resistant genes [21,22,23,24]. At present, researchers have developed advanced oxidation, adsorption, microbial degradation and other technologies for the treatment of TC, among which photocatalytic oxidation is a typical advanced oxidation method [25,26,27]. In photocatalytic degradation, photocatalysts absorb solar energy to degrade antibiotics [28,29]. Owing to its environmental protection, low cost, complete degradation, and mild reaction conditions, this technology has become a research hotspot at home and abroad [30]. Nanoporous TiO_2_ usually has a high specific surface area, pore diameter, and isotropic characteristics, can fully contact with TC solutions and has good reproducibility capacity, so it has been the most studied and widely used materials [31].

Herein, we report a facile one-step method to synthesize the hierarchical micro/nanostructures of nanoporous TiO_2_ with hierarchical micro/nanostructures under mild reaction conditions and free of acids and bases. This method not only has a facile operation and low solvothermal reaction temperature, but also has excellent photocatalytic performance for tetracycline under visible light irradiation. This facile method for controllable synthesis of nanoporous TiO_2_ with hierarchical micro/nanostructures is expected to become a new low-cost industrial production method.

## 2. Materials and Methods

### 2.1. Materials

Cetyltrimethylammonium bromide (CTAB), ethanol, tetrabutyl titanate (TBOT), tetracycline, and commercial TiO_2_ (P25) were obtained from Macklin Biochemical Co., Ltd., Shanghai, China. All these chemical reagents were used as received without further purification.

### 2.2. Preparation of Nano-TiO_2_

In this experiment, the hierarchical micro/nanostructures of nanoporous TiO_2_ were synthesized by a one-step low-temperature solvothermal method, and the morphology and photocatalytic degradation of TiO_2_ were controlled by adjusting the solvothermal reaction temperature. In a typical process, as illustrated in Figure 1a, 0.72 g of CTAB was added to 50 mL ethanol. After full mixing, 6 mL of TBOT was slowly added through the pipette and stirred for 1 h. The mixed solution was then transferred into a Teflon-lined stainless steel autoclave with a capacity of 100 mL. Subsequently, the autoclave was heated at 100 °C for 24 h and then cooled down to an ambient temperature naturally. Next, the as-obtained product was centrifuged, washed, and redispersed in water and ethanol for three cycles, respectively. After that, the product was dried at 80 °C in an oven for 12 h and then calcined at 500 °C in the air for 2 h. Finally, the product was named TiO_2_-100. For comparison, TiO_2_-90 and TiO_2_-110 were also prepared using the same procedures under reaction temperatures of 90 °C and 110 °C, respectively. Figure 1b shows the large-scale synthesis of TiO_2_-100 obtained by magnifying 5 times according to the typical experiment, with a total mass of about 0.011 Kg.

### 2.3. Characterization

The crystalline structures were characterized by an X-ray diffractometer (XRD, Ultima IV, Rigaku, Tokyo, Japan) with Cu Kα radiation at a scanning rate of 10°/min in the 2θ ranges of 10–80°. The field emission scanning electron microscopy (FESEM) images were obtained by S-4800 (Hitachi, Tokyo, Japan) equipment. The measurements based on a transmission electron microscope (TEM) were carried out on JEM-2010HR (JEOL, Tokyo, Japan). The N_2_ adsorption measurements were conducted by an ASAP 2010 analyzer (Micromeritics, Norcross, GA, USA). The Barrett–Joyner–Halenda (BJH) pore diameter distribution curves were obtained from the desorption branch, and specific surface areas were obtained according to the Brunauer–Emmett–Teller (BET) model. The Fourier transform infrared spectroscopy (FTIR, Nicolet 6700, Thermo Scientific, Waltham, MA, USA) of samples was performed over the wavenumber range of 500–4000 cm^−1^, of which the resolution is 4 cm^−^^1^, and the scan mode was transmittance. The Raman spectra were measured by a Raman spectrometer (DXR 2, Thermo Scientific, Waltham, MA, USA), with a 785 nm He-Ne laser and scanned within 50–1000 cm^−^^1^.The chemical characterization of the sample surface was performed using an X-ray photoelectron spectroscopy (XPS, ESCALAB 250Xi, Thermo Scientific, Waltham, MA, USA). The electron paramagnetic resonance (EPR) spectra were recorded at room temperature (Bruker EMXplus, Karlsruhe, Germany). The ultraviolet-visible diffuse reflectance spectrum (UV–vis DRS) was obtained using a UV-Vis spectrometer (Shimadzu, UV-270, Kyoto, Japan). The photoluminescence (PL) spectra were recorded on an FLS980 spectrophotometer (Edinburgh, UK); a few powder samples were placed in the sample table, the excitation (EX) wavelength was set to 320 nm, the excitation slit and the emission slit were both 10 nm and the test voltage is 400 V. TC solutions irradiated with UV light for different times were collected, and the intermediate compounds formed during the degradation of TC were analyzed by liquid chromatograph mass spectrometry (HPLC-MS, Thermo Scientific, Waltham, MA, USA). The analysis was employed to identify intermediate compounds formed during TC degradation. The transient photocurrent (TPC) was measured by Electrochemical Station (Chenhua, CHI 660C, Shanghai, China), the standard three-electrode system was used in the measured process, the electrolyte was 0.01 M Na_2_SO_4_ solution, and the light source was 300 W Xe-lamp (λ > 300 nm).

### 2.4. Photocatalytic Degradation of TC 

First, 50 mg of photocatalyst and tetracycline hydrochloride solutions (100 mL, 20 mg/L) were mixed for 30 min in the dark to ensure complete adsorption–desorption equilibrium. The ozone-free Xe arc lamp (HXF 300 W, 300 W) attached with a UV cut filter (λ > 420 nm) was used as the visible light source. The distance between the reactor and the light was fixed at 15 cm, the light irradiation intensity was controlled to 60 mW/cm^2^, and the reactor temperature was kept at room temperature by recirculating cooling water. After reaching the equilibrium, the photodegradation experiment was carried out under the irradiation of the Xe lamp, and 3 mL of aliquot suspension was taken at regular 10 min intervals. The obtained sample solution was filtered through a 0.22 μm membrane and the concentration of TC solution was analyzed by a UV-visible spectrophotometer.

## 3. Results and Discussion

### 3.1. XRD Analysis 

Due to the fact that the anatase phase of TiO_2_ has a higher density of localized states and surface-adsorbed hydroxyl groups, the photocatalytic performance of TiO_2_ in the anatase phase is generally better than that in the rutile phase. Hence, the focus of this study is mainly placed on the anatase phase of TiO_2_. The calcination temperature is an important factor that affects the phase transition and the morphology of TiO_2_ [32]. To obtain a highly crystalline anatase phase of TiO_2_, we first explored its phase transition temperature by in-situ XRD. The in-situ XRD pattern is shown in Figure 2. At the calcination temperature of 400 °C, the (101) crystal plane formed firstly in the anatase phase of TiO_2_ (JCPDS No. 21-1272). At this time, the diffraction peak of TiO_2_ was large and wide, and the crystallinity was low. When the temperature rose to 500 °C, all the characteristic peaks of TiO_2_ sample in the anatase phase appeared, and the 2θ value formed (103), (004), and (112) crystal planes around 38°. When the temperature rose to 600 °C, TiO_2_ samples presented the most significant characteristic peak in the anatase phase. At this time, the crystallinity of TiO_2_ was relatively high, and all of them were in anatase phases. When the temperature further rose to 700 °C, the (110) crystal plane began to appear in the rutile phase of TiO_2_ samples (JCPDS No. 21-1276), while the characteristic diffraction peak of the (101) crystal plane in the anatase phase began to weaken. This indicated that some TiO_2_ powder was transformed from the anatase phase to the rutile phase, which confirmed that the phase transition temperature of TiO_2_ was less than 700 °C. As the temperature further rose, increasing TiO_2_ samples were transformed from the anatase phase to the rutile phase. When the temperature reached 950 °C, all the characteristic peaks of TiO_2_ in the anatase phase disappeared, and the characteristic peaks of TiO_2_ in the rutile phase were significant and sharp. This indicated that the rutile phase of all TiO_2_ samples was formed, with high crystallinity.

The XRD patterns of the synthesized materials after calcination at 500 °C for 2 h are shown in Figure 3. It can be seen the 2θ values of TiO_2_-100 and TiO_2_-110 were 25.3°, 36.9°, 37.8°, 38.6°, 48.0°, 53.9°, and 55.0°, which had significant characteristic diffraction peaks, corresponding to (101), (103), (004), (112), (200), (105) and (211) crystal planes of TiO_2_ in the anatase phase (JCPDS No.21-1272), respectively [33]. However, TiO_2_-90 has only one characteristic peak around 38° at the 2θ value, indicating that when the solvothermal reaction temperature was 90 °C, the synthesized TiO_2_ was not completely in the anatase phase. These characteristic diffraction peaks of TiO_2_-100 were sharp and narrow, which indicated that TiO_2_-100 had high crystallinity. No other significant diffraction peaks were found in the XRD spectrum, indicating that the prepared TiO_2_ had high purity. In addition, the crystallite size was calculated using the Debye–Scherrer equation as follows [34]:(1)$$D=\frac{K\lambda }{\mathrm{\beta cos}\theta }\;\;\;\;\;$$
where *D* is the nanoparticle crystalline diameter, *K* represents the Scherrer constant and equals 0.89, *λ* is the X-ray wavelength (1.54 Å for copper K-α), and *β* denotes the full width at half maximum (FWHM). The average crystallite diameters of TiO_2_-90, TiO_2_-100, and TiO_2_-110 particles of 9.3 nm, 14.6 nm, and 18.1 nm were obtained from the analysis of the main reflections, respectively.

### 3.2. Morphologic Structure Analysis

The morphology and surface nanoparticle diameter distribution (insert) of nanoporous TiO_2_ with hierarchical micro/nanostructures are shown in Figure 4. It can be seen from Figure 4a–d that the morphology of the synthesized TiO_2_ was spherical with a rough surface, and the surface was composed of numerous hierarchical nanocrystallites (size: 5–22 nm). As the reaction temperature rose, the spherical diameter of samples gradually increased, and the nanoparticles on the surface became larger and denser. The diameter of TiO_2_-90, TiO_2_-100, and TiO_2_-110 was about 300 nm (Figure 4a), 320 nm (Figure 4b), and 360 nm (Figure 4c), respectively. When the reaction temperature further rose at 130 °C, the diameter of TiO_2_-130 reached 500 nm (Figure 4d). This indicates the size of nano-porous TiO_2_ can be controlled by controlling the reaction temperature. This special hierarchical micro/nanostructure of nano-porous TiO_2_ is beneficial to the separation of photogenerated electron pairs and improve the photocatalytic performance [1]. The elemental mapping shows that TiO_2_ only has Ti and O elements, which is consistent with XRD analysis results (Appendix A).

Figure 4e,f present TEM and HRTEM images of TiO_2_-100. It can be seen from Figure 4e that TiO_2_-100 exhibited a solid spherical shape with a relatively uniform size. From the HRTEM image of Figure 4f, it can be seen that the TiO_2_-100 lattice stripes were clear, and the lattice spacing was 0.352 nm, which corresponded to the (101) crystal plane of TiO_2_ in the anatase phase (JCPDS No. 21-1272).

### 3.3. Surface Area and Pore Structure

The BET-specific surface area and corresponding pore size distribution of synthetic TiO_2_ were analyzed by the N_2_ adsorption–desorption isotherm curve, as shown in Figure 5. Based on the IUPAC classification, the N_2_ adsorption–desorption isotherms of all synthesized samples can be classified into type IV adsorption isotherms, which correspond to ordered mesopores TiO_2_ [35]. Among them, TiO_2_-100 exhibited the H_2_ type hysteresis loop, indicating that the pore size distribution of mesopores in the material was relatively uniform. In contrast, TiO_2_-90 and TiO_2_-110 exhibited the H_3_ type hysteresis loop, indicating that the mesoporous structure of these materials was irregular [36,37] (Figure 5a). By comparing the specific surface area of these samples, it can be seen that the specific surface area (57 cm^2^/g) of TiO_2_-100 was much larger than that of TiO_2_-90 and TiO_2_-110, and it was 6.2 times that of commercial P25. The pore size distribution curve calculated by the BJH method further revealed that TiO_2_-100 had a mesoporous structure, and the particle size distribution was relatively concentrated, as shown in Figure 5b. The BET-specific surface area, pore volume, and average pore diameter of these synthesized samples are listed in Appendix A. Compared with commercial P25, TiO_2_-100 exhibits a larger specific surface area and pore volume. The favorable specific surface area and internal pore structure are beneficial to the multiple reflection and refraction of light, thus boosting the utilization efficiency of light and raising the photocatalytic ability [38,39].

### 3.4. FTIR and Raman Spectra

The surface chemical functional groups of TiO_2_-100 samples were detected by FTIR spectra, with the results shown in Figure 6a. The peaks around the wavenumbers of 3434 cm^−1^ and 1631 cm^−1^ corresponded to the stretching and bending vibration peaks of the -OH group, which was the peak of adsorbed water on the sample surface [40,41,42], and they corresponded to the stretching vibration peaks of the O-Ti-O bond at 600 cm^−1^. This indicated that TiO_2_-100 had abundant hydroxyl functional groups and the presence of oxygen vacancy, which may provide more active sites for photocatalytic reactions [43,44,45].

The structure information of TiO_2_-100 was further investigated by Raman spectra. From the Raman spectrum in Figure 6b, it can be seen that there were five typical Raman peaks at 145, 197, 398, 518, and 639 cm^−1^ within the range of 150–1000 cm^−1^, which represented Ti-O single bond stretch vibration peaks in TiO_2_-100 crystal in the anatase phase, which is consistent with the previous reports by Maqbool et al. [46,47]. This finding was in agreement with the XRD analysis results.

### 3.5. Composition and Valence State Analysis

The distribution and valence state of elements on the surface of photocatalytic materials were explored by XPS technology, as shown in Figure 7. The survey XPS spectrum of TiO_2_-100 is shown in Figure 7a. The characteristic peaks of C 1s, Ti 2p, and O 1s can be observed from the spectrum, which indicates that the prepared sample contained C, O, and Ti elements. The C element may come from CO_2_ or C pollution adsorbed in the air (Figure 7b). Figure 7c presents the XPS spectrum of Ti 2p, when the FWHM of the deconvolution of the spectra was fixed at 1.1 eV, the percentages for Ti^4+^ 2p_1/2_, Ti^4+^ 2p_3/2_, Ti^3+^ 2p_1/2_ and Ti^3+^ 2p_3/2_ are 15.24%, 59.86%, 21.47% and 3.43%, respectively. It can be seen from the spectrum that the binding energies at 463.2 eV and 457.9 eV corresponded to Ti^4+^ 2p_1/2_ and Ti^4+^ 2p_3/2_, which corresponded to the Ti^4+^-O bond in TiO_2_ [48]. Similarly, the peaks at 464.0 eV and 458.5 eV binding energies corresponded to Ti^3+^ 2p_1/2_ and Ti^3+^ 2p_3/2_. The main function of Ti^3+^ was to balance the charge surplus caused by oxygen vacancies. In the high-resolution O1s diagram (Figure 7d), it can be observed that the oxygen spectrum had the main peaks with binding energies at 529.9 eV and 531.6 eV, which corresponded to the characteristic peaks of the Ti-O bond and surface hydroxyl oxygen (O_OH_), respectively [49,50,51,52,53].

### 3.6. EPR Spectra

We used the EPR characterization technology to further study the reactive oxides and oxygen vacancy generated by the synthetic samples under light, and the EPR spectrum is shown in Figure 8. Figure 8a,b are, respectively, the EPR spectra of hydroxyl radicals and superoxide radicals generated by the samples after irradiation with visible light for 1 min. Figure 8a,b show that all the synthesized samples produced hydroxyl radicals and superoxide radicals after irradiation, which are the main active species for TiO_2_ to degrade pollutants. Under the same test parameters, TiO_2_-100 has higher signal intensity than TiO_2_-90 and TiO_2_-110, indicating that TiO_2_-100 generates more hydroxyl radicals and superoxide radicals and is more efficient in photocatalysis [54]. Figure 8c is the oxygen vacancy EPR spectrum of the synthetic samples. It can be seen from the results that all samples have significant characteristic peaks at the position where the g value is 2.003, which indicates the existence of oxygen vacancies in the samples. These may be due to the combination of Ti^3+^ on the surface of the sample and oxygen in the air or water, and the existence of oxygen vacancies is beneficial to the oxidation and reduction processes in photocatalytic reactions. This is consistent with the XPS analysis results. Moreover, the oxygen vacancy signal intensity of TiO_2_-100 is stronger than that of TiO_2_-90 and TiO_2_-110, indicating that TiO_2_-100 may have more oxygen vacancies [55,56].

### 3.7. Optical Absorption, Photoluminescence, and Photoelectrochemical Property

The UV–vis diffuse reflectance spectra of catalysts were examined to evaluate the optical properties of these samples, with the results shown in Figure 9a,b. It can be seen from Figure 9a that the samples prepared at different reaction temperatures and P25 had similar absorption spectra. Both in the ultraviolet region (λ < 400 nm) had strong absorption, which was mainly attributed to the band gap (3.2 eV) excitation of TiO_2_ in the anatase phase. As shown in Figure 9b, it can be calculated based on the Kubelka Munk function [1] that the band gap of these TiO_2_ samples was about 3.08 ev, and that of P25 was about 3.12 ev [25,30,57].

The electron–hole pair migration and transfer efficiency of the catalyst were investigated by photoluminescence (PL) emission spectroscopy. The PL spectrum demonstrated that the separation efficiency of electron–hole pairs was lower when the PL emission peak intensity was higher [40,58]. It can be seen from Figure 9c that under the excitation of 320 nm, all synthesized materials and P25 exhibited high-intensity fluorescence peaks located at 410 and 469 nm, which, respectively, result from self-trapped excitons in bandgap transition and electrons trapped by oxygen vacancies. According to literature reports, the presence of oxygen vacancies allows achieving efficient carrier transport and increasing the carrier concentration [56]. Among them, TiO_2_-110 exhibited the highest intensity, while TiO_2_-100 had a low emission intensity. This result indicated that the electron–hole pairs of TiO_2_-100 were the lowest relative to the other samples.

Figure 9d shows the transient photocurrent response curves of different samples. Generally, the stronger the photocurrent is, the better the photogenerated charge carriers of the catalyst can capture light, and the higher the separation efficiency [59,60]. During 50 s of irradiation, all samples exhibited a fast and stable transient photocurrent response. The photocurrent intensity can be ranked as follows: TiO_2_-100 > TiO_2_-110 > TiO_2_-90 > P25. The current density of TiO_2_-100 can reach 60 μA/cm^2^, which increased by 10 times compared with P25 (6 μA/cm^2^). This result suggested that the TiO_2_-100 can effectively transfer and separate photogenerated carriers, thus improving the photocatalytic performance.

### 3.8. Photocatalytic Degradation and Mechanisms of TC

The photocatalytic activity of the catalyst to degrade antibiotic TC was evaluated under UV and visible light irradiation, with the results shown in Figure 10. It can be seen from Figure 10a that under UV-light irradiation, in the absence of catalysts, the TC concentration remained unchanged with the increase of irradiation time. This result indicated that the photolysis of TC was negligible in the absence of TiO_2_. Additionally, before the irradiation under a light source, all of these synthesized TiO_2_ samples could adsorb TC in the dark within 30 min, and the adsorption rate was about 20%, which was similar to the research result of Wu [21]. Moreover, under the irradiation of simulated sunlight, the degradation of TiO_2_-100 to TC was the fastest and most thorough, and TC could be completely removed within 30 min. The TC degradation rate of TiO_2_-90 and TiO_2_-110 was also nearly 100% after 50 min, while that of P25 only reached 54% after 60 min under light irradiation, which was in stark contrast to the degradation efficiency of TiO_2_-100.

We further explored the ability of catalysts to degrade TC under visible light, as shown in Figure 10b. Under visible light irradiation, in the absence of catalysts, the TC concentration remained unchanged with the increase of irradiation time. After 60 min of visible light irradiation, the degradation rate of TiO_2_-100 to TC can reach 79%, which was much better than that of TiO_2_-90 (74%) and TiO_2_-110 (71%). In contrast, under the same conditions, the degradation rate of P25 to TC within 60 min was only 29%, which was only 13% higher than that of the blank experiment with only light in the absence of catalysts. Since TiO_2_ can absorb only UV light, the visible light degradation of TC may be attributed to the TC effectively sensitizing TiO_2_, in which TC forms a visible-light responsive TC-TiO_2_ complex via a ligand-to-metal charge transfer mechanism. The degree of sensitization was determined by the formation of a visible light-responsive TC-TiO_2_ complex via surface adsorption and the overlap of HOMO/LOMO of TC and the conduction band of TiO_2_. Furthermore, ·O_2_^−^ species played a critical role during the visible light photocatalytic degradation of TC by TiO_2_ [61].

Figure 10c,d present the kinetic fitting results of the degradation of TC by TiO_2_ under UV light and visible light, respectively. The results indicated that the photocatalytic degradation process of TC by TiO_2_ was consistent with pseudo-first-order kinetics. The pseudo-first-order kinetic rate constants of different samples are listed Appendix A. Here, when under the condition of ultraviolet light, the rate constant of TiO_2_-100 can reach 0.185 min^−1,^ which was 2, 1.8, and 19 times that of TiO_2_-90(0.090 min^−1^), TiO_2_-110 (0.105 min^−1^), and P25 (0.009 min^−1^), respectively. Furthermore, under visible light irradiation, the K value of TiO_2_-100 can reach 0.217 min^−1^, which was 12, 11, and 43 times that of TiO_2_-90 (0.016 min^−1^), TiO_2_-110 (0.018 min^−1^) and P25 (0.005 min^−1^), respectively. This indicated that the synthesized TiO_2_-100 had a stronger degradation ability than P25, and it was comparable to the degradation rate constants of TC by the TiO_2_-graphitized carbon composite prepared by Zhang [62].

Appendix A summarizes some of the studies on the degradation of TC by TiO_2_ catalysts [32,53,61,63,64,65,66,67,68]. It can be concluded from Appendix A that as-prepared TiO_2_ nano-particles can efficiently remove TC compared to various photocatalysts.

The performance stability of the catalyst determines its potential application value. To further investigate the degradation performance stability of the samples, the degradation performance of TiO_2_ on TC was studied for four cycles under UV-light and visible light irradiation, as shown in Appendix A. After four cycle experiments, the degradation efficiency of TC by TiO_2_-100 was only reduced by about 10%. The fourth degradation efficiency was 93% (UV-light) and 70% (visible light), respectively, indicating that it has good stability. After four cycles, the possible reasons for the decrease in catalytic activity are the incomplete recovery of powder TiO_2_-100 during the recovery process and insufficient cleaning of TC adsorbed on TiO_2_-100 active sites [63].

### 3.9. Intermediates and Possible Degradation Pathways of TC

To further clarify the photodegradation pathway and mechanism of TC by TiO_2_, the degradation process of TC was predicted by the density functional theory (DFT) calculation. Based on density functional theory calculations, the Gaussian 16 program software and the Multiwfn program were used to perform abbreviated Fukui function calculations to reveal the reactivity between different atomic sites on the TC molecule and the dominant active oxide species in the TiO_2_ system [27,69,70]. First, Gaussian was used to optimize the structure of TC molecules (as shown in Figure 11a), and then the natural bond orbital (NBO) analysis was performed to calculate the single-point energy to obtain the NBO charge distribution of each atom of the optimized TC molecule under the disturbance of external electrons. Subsequently, the Multiwfn program was adopted to perform the conversion into a Hirshfeld chare, so as to calculate the abbreviated Fukui function indices corresponding to electrophilic, nucleophilic, and free radical attacks, respectively, as shown in Appendix A. The values marked in color represented the top 10 atoms of electrophilic (f^−^), nucleophilic (f^+^), and free radical (f^0^) Fukui function indices on the TC molecule [71].

It can be seen from Appendix A that the f^−^ and f^0^ values of O7 were the largest. Hence, it was more vulnerable to being preferentially attacked by the electrophile substances (^1^O_2_,h^+^) and free radicals (·OH). Additionally, the f^+^ value of O6 was larger, and it was most likely to be attacked by the nucleophiles substances (·O_2_^−^), leading to nucleophilic reaction. However, those sites at O4, O6, C15, and C27 may be preferentially attacked by electrophiles, nucleophilic, and free radicals at the same time, as shown in the yellow background of Figure 11b–d.

The intermediates formed during the degradation of TC were analyzed using HPLC-MS. Figure 12 presents the HPLC-MS spectrum recorded by TC in the TiO_2_-100 system under different UV irradiation time lengths. It can be seen from the figure that when the residence time (RT) was 10.4 min, the peak corresponding to the specific charge (*m*/*z*) of TC at 445 gradually decreased, and basically disappeared after 30 min. This finding was consistent with the results obtained in the analysis of Figure 10a. At the same time, a large number of new small molecule fragments appeared, and they were all intermediates formed in the degradation process. After identification, the possible intermediates were named P1–P10 and listed in Appendix A.

Through the calculation of the above Fukui function, the carbon–carbon double bond, phenolic hydroxyl group, and amino group of the TC molecule can be revealed [43,46,63,68]. These high-electron density structures will be preferentially attacked by active oxide species. Combining the degradation intermediates detected by LC-MS and the results of theoretical calculations, we proposed two possible degradation pathways of TC in the TiO_2_-100 reaction system, as shown in Figure 13. In terms of the first pathway on the left, TC was first attacked by reactive species and lost two methyl groups to form intermediates P1 (*m*/*z* = 417). In terms of the second path, the carbon–carbon double bond (C28) of TC was attacked by the nucleophile substances to form an intermediate P2 (*m*/*z* = 462) through hydroxylation. The intermediate P2 can form P3 (*m*/*z* = 434) through demethylation under the action of active oxidative species. The intermediates P1 and P3 can also form P4 (*m*/*z* = 402) and P5 (*m*/*z* = 385) through deamidation. The O1 and O4 sites of P4 and P5 were attacked by active components, and the carbon–carbon double bonds at C15 and C20 sites were also destroyed to generate intermediates P6 (*m*/*z* = 340). Active species further attacked the C16, C25, and C31 sites for dehydroxylation to form intermediates P7 (*m*/*z* = 289). Subsequently, the carbon–carbon double bonds at C16, C17, C26, and C27 were attacked by active oxide species and to generate a carboxyl group through the ring-opening reaction, thus forming P8 (*m*/*z* = 277). Finally, it generated P9 (*m*/*z* = 60) and P10 (*m*/*z* = 118) after decarboxylation. These intermediates will eventually be mineralized into small molecule harmless products such as CO_2_ and H_2_O [27,43,62].

## 4. Conclusions

In this study, the hierarchical micro/nanostructures of nanoporous TiO_2_ were prepared on a large scale at a low temperature by a facile one-step solvothermal process. This method not only enhanced the visible light response of the catalysts and promoted the separation of photogenerated electron pairs but also exhibited excellent photodegradation performance for TC under visible light irradiation. In this study, a facile and energy-saving new synthesis method was proposed for the industrial production of nanoporous TiO_2_ with hierarchical micro/nanostructures and reducing the synthesis temperature. These findings are expected to provide experimental and theoretical possibilities for the photocatalytic degradation of antibiotics. The main conclusions can be elucidated as follows.

(1) These synthetic catalyst materials were analyzed by XRD, SEM, BET, and XPS, and all the analyses verified that the catalyst synthesized by the solvothermal reaction temperature at 100 °C and calcination at 500 °C presented a favorable structure, morphology, and optical properties.

(2) The synthesized catalysts exhibited excellent photocatalytic degradation performance for antibiotic TC, regardless of the irradiation of UV light or visible light. The degradation rate of TiO_2_-100 to TC was close to 100% (UV light) and 79% (visible light), and its photocatalytic activity was better than that of P25. This was attributed to the TC-sensitizing of TiO_2_ to form a complex for visible light absorption.

(3) Two possible photocatalytic degradation pathways of TC and possible intermediate products were proposed by the density functional theory calculation and HPLC-MS detection. The degradation process of TC can be mainly attributed to the breakage and rearrangement of carbon–carbon double bonds, nitrogen-carbon bonds, and hydroxyl groups after being attacked by active species. They were finally mineralized into nontoxic small molecule products, such as CO_2_ and H_2_O.

In summary, this work not only provides a potential approach in the exploration of nanoporous TiO_2_ materials, but also provides a theoretical basis and technical support for the efficient degradation of antibiotics.

## Figures and Tables

**Figure 1 nanomaterials-14-00943-f001:**
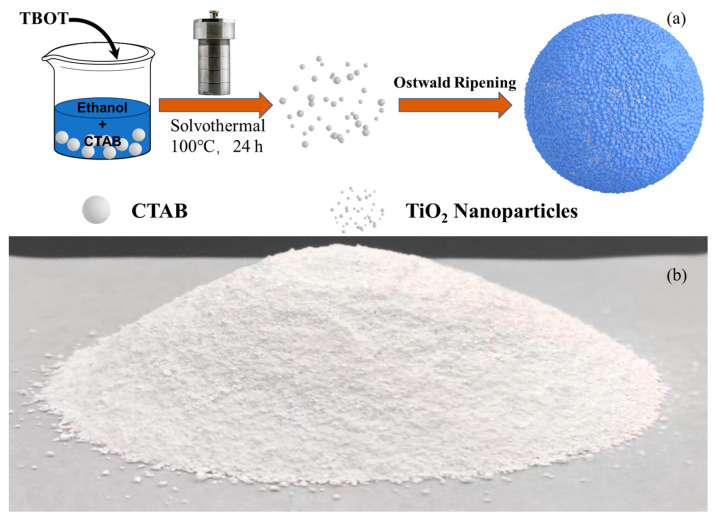
(**a**) Illustration of the preparation of TiO_2_-100; (**b**) The nanoporous TiO_2_-100 were synthesized on a scale of 5 times.

**Figure 2 nanomaterials-14-00943-f002:**
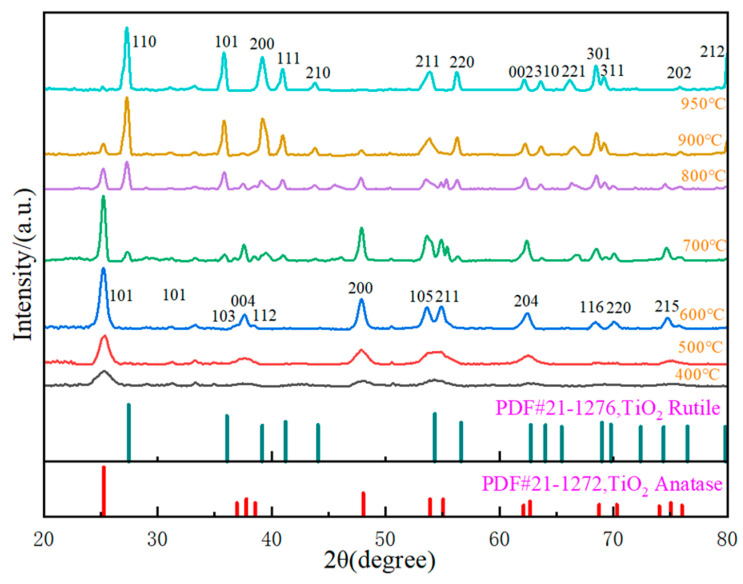
The patterns of in situ XRD at different temperatures.

**Figure 3 nanomaterials-14-00943-f003:**
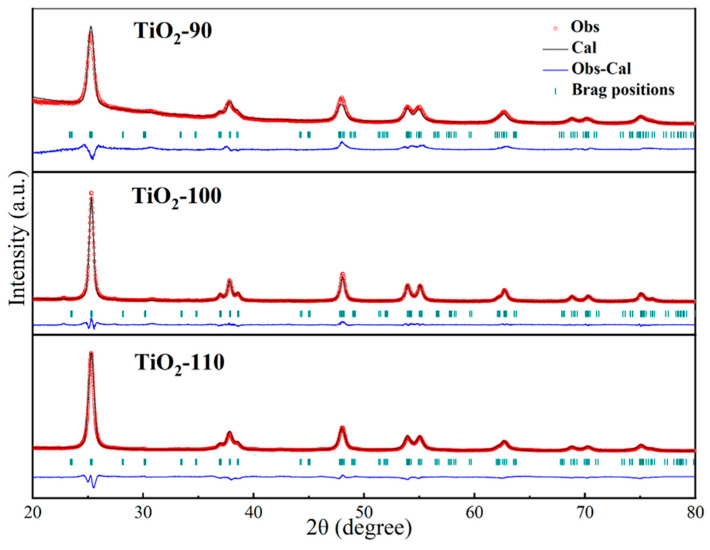
Rietveld refinement of samples at different solvothermal temperatures.

**Figure 4 nanomaterials-14-00943-f004:**
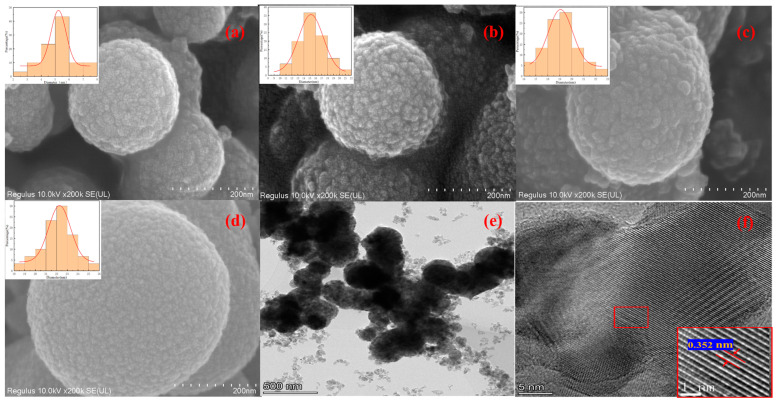
SEM images and surface nanoparticle diameter distribution(insert) of (**a**) TiO_2_-90; (**b**) TiO_2_-100; (**c**) TiO_2_-110; (**d**) TiO_2_-130; (**e**) TEM image of TiO_2_-100; (**f**) HRTEM images of TiO_2_-100.

**Figure 5 nanomaterials-14-00943-f005:**
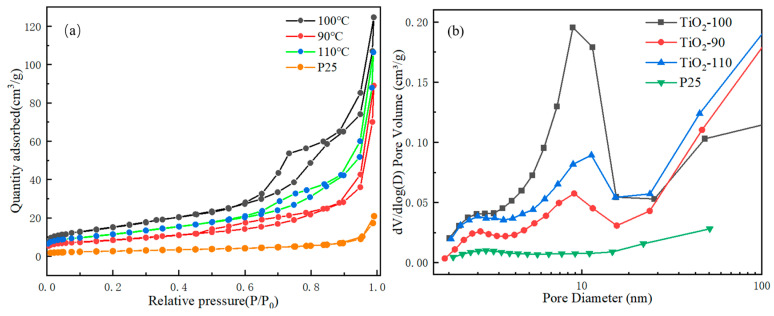
(**a**) N_2_ adsorption–desorption isotherm and (**b**) the pore size distribution curves of synthesized samples.

**Figure 6 nanomaterials-14-00943-f006:**
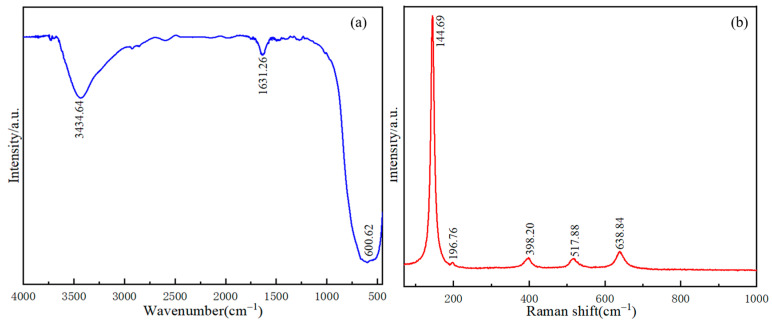
(**a**) The FTIR spectra of TiO_2_-100; (**b**) Raman spectra of TiO_2_-100.

**Figure 7 nanomaterials-14-00943-f007:**
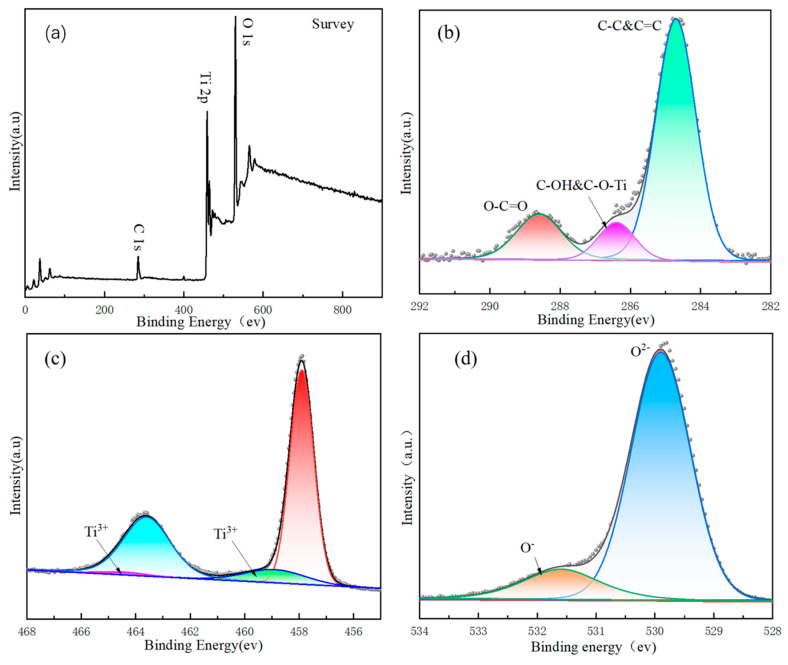
XPS spectra of TiO_2_-100 (**a**) survey, (**b**) C1s, (**c**) Ti 2p, and (**d**) O 1s.

**Figure 8 nanomaterials-14-00943-f008:**
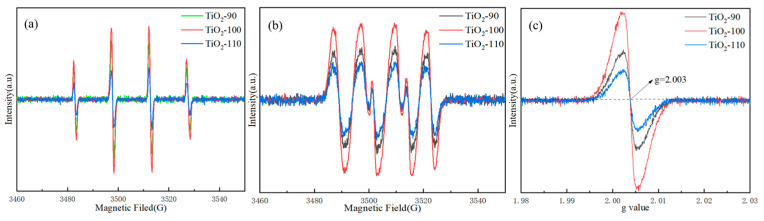
The EPR spectra of samples (**a**) hydroxyl radical; (**b**) superoxide radical; (**c**) oxygen vacancy.

**Figure 9 nanomaterials-14-00943-f009:**
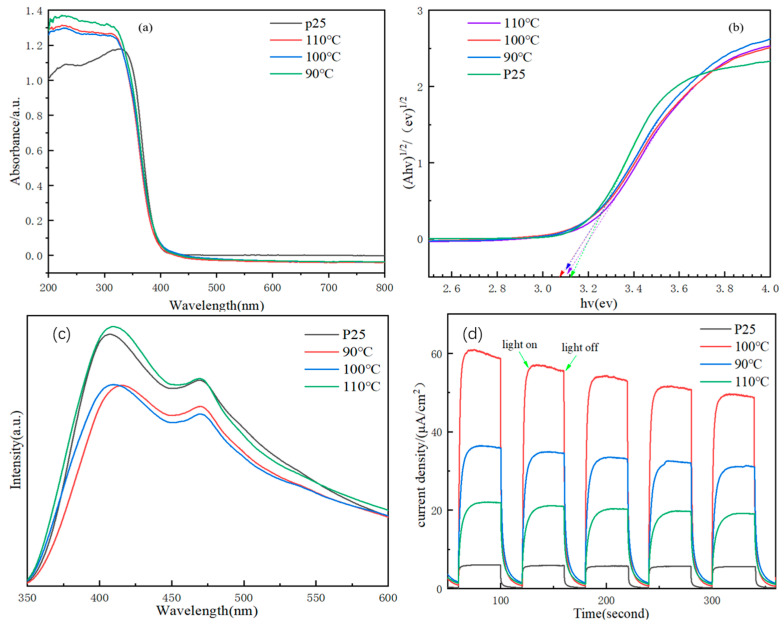
(**a**) UV–vis diffuse reflectance spectra; (**b**) The plot of (αhν)^1/2^ versus hν; (**c**) Photoluminescence spectra; (**d**) Transient photocurrent response spectra of photocatalysts.

**Figure 10 nanomaterials-14-00943-f010:**
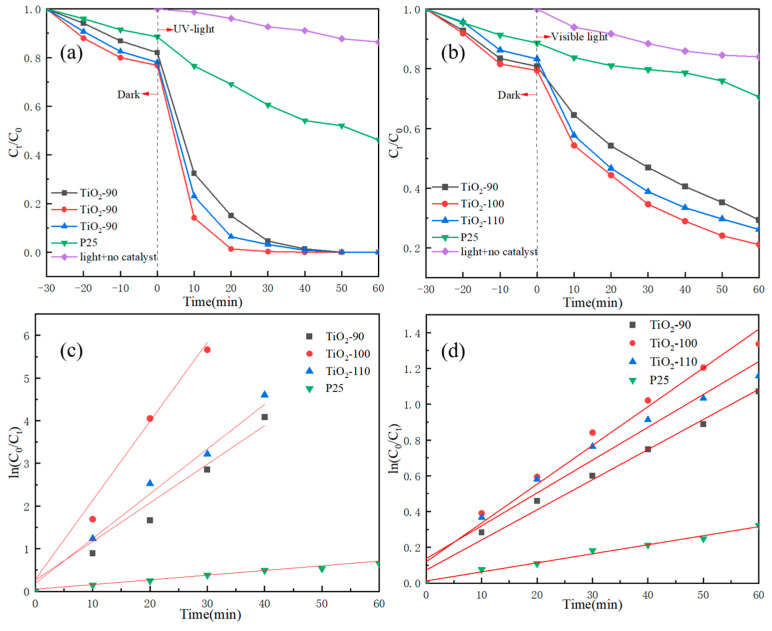
TC photodegradation over TiO_2_: (**a**) photodegradation efficiency of UV-light; (**b**) photodegradation efficiency of visible light; (**c**) reaction kinetics of UV-light; (**d**) reaction kinetics of visible light.

**Figure 11 nanomaterials-14-00943-f011:**
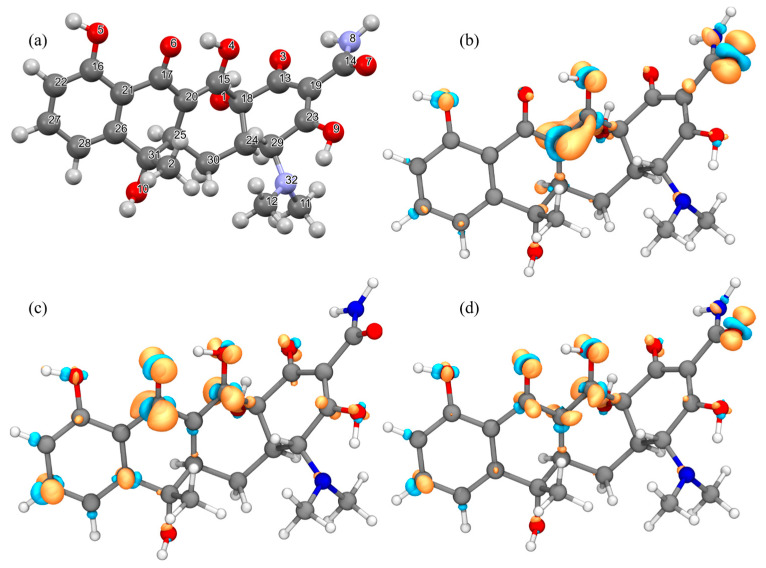
(**a**) Optimized structure of TC molecular; (**b**) The sites prone to electrophilic reactions; (**c**) The sites prone to nucleophilic reactions; (**d**) The sites prone to free radical reactions.

**Figure 12 nanomaterials-14-00943-f012:**
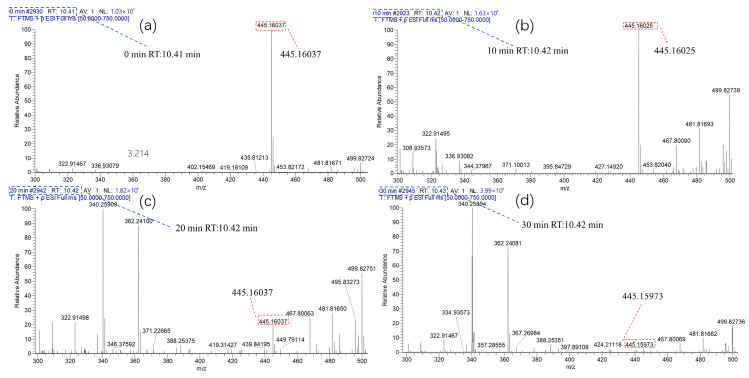
HPLC-MC spectra of TiO_2_-100 with different degradation time lengths of TC under UV-light irradiation: (**a**) 0 min; (**b**) 10 min; (**c**) 20 min; (**d**) 30 min.

**Figure 13 nanomaterials-14-00943-f013:**
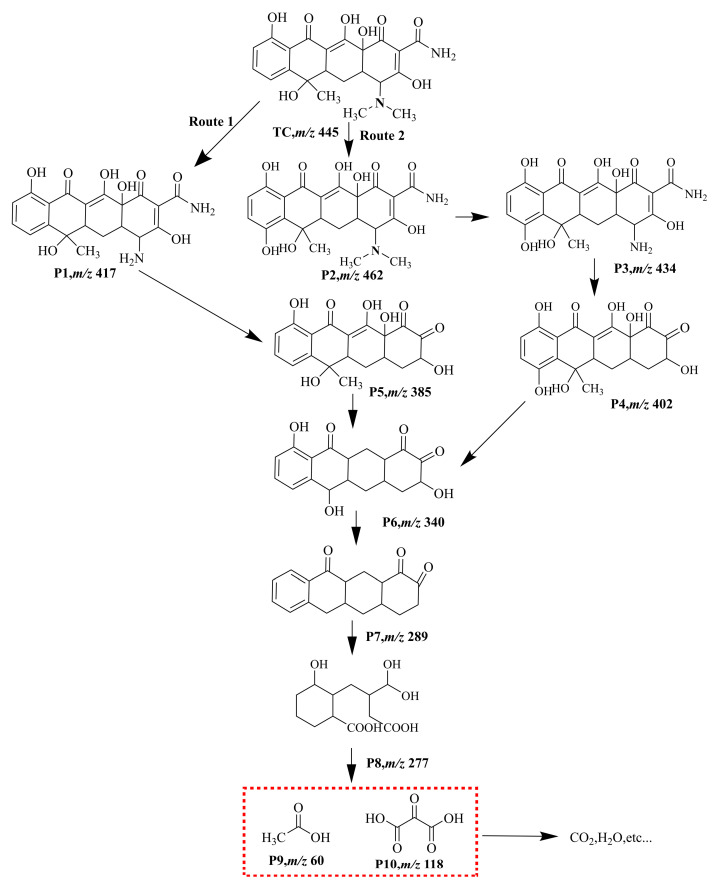
The possible photocatalytic degradation pathway of TC by TiO_2_-100.

## Data Availability

The data are available on the request from the corresponding authors.

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
