# Peer review of "Facile Synthesis to Porous TiO2 Nanostructures at Low Temperature for Efficient Visible-Light Degradation of Tetracycline"

_nanomaterials, 2024, doi:10.3390/nano14110943_

Round 1

Reviewer 1 Report

Comments and Suggestions for Authors

In this work, the author synthesized nanoporous TiO2 with hierarchical micro/nanostructures, demonstrating high efficiency in degrading tetracycline under both UV and visible light. In visible light irradiation, the degradation rate was 79% after 60 minutes, compared to TiO2, which achieved only 29% photodegradation. The study also explored possible intermediates formed during the photocatalytic degradation of TC using density functional theory calculations and HPLC-MS spectra. Overall, this research introduces a novel approach for hierarchical micro/nanostructured nanoporous TiO2, with potential applications in industrial manufacturing. However, substantial revision is needed before acceptance for publication.

Introduction section: (1) Comparison with other reported methods is necessary to understand the advantages of the solvothermal method. The novelty aspect requires clarification. (2) An explanation is needed on how the use of powder dispersion like TiO2 NPs is advantageous for removing TC from water. (3) The reproducibility capacity of the catalysts used should be addressed. (4) The author quoted a specific study .. “it was also confirmed that the degradation rate of catalysts to rhodamine G was relatively low (60%)”. .This is not sufficient to draft the claim that the TiO2 from the literature is an incompetent catalyst. Please revise.

Methods section: (1) A description of how the samples were prepared for photoluminescence (PL) analysis is needed. (2) Details on the intensity of light, distance, and effect of temperature on the Xe arc lamp (HXF 300W, 300 W) should be provided.

Results and Discussion: (1) Baseline correction of XRD spectra reported in Figure 2 is necessary. (2) Refinement of the XRD spectra (Fig. 2) is required to precisely determine the TiO2 phase. (3) FTIR, Raman, and XPS sections need revision in terms of discussion from literature, including recent studies on Anatase TiO2, such as 10.1021/acscatal.3c06203. (4) Application of kinetics to the photocatalytic data reported in fig. 9 would be useful. (5) An explanation is needed on how samples were collected for HPLC analysis for TC-intermediates analysis. (6) Figure 11 resolution needs improvement. Please revise.

Conclusion: (1) Future perspectives should be included.

Comments on the Quality of English Language

Acceptable, however, can be further improved.

Reviewer 2 Report

Comments and Suggestions for Authors

This manuscript reports a facile one-step method to synthesize TiO2 nanoporous structures and evaluate its application in photocatalytic degradation of TC drug under visible light irradiation. The approach can be moderately upscaled. The materials are thoroughly characterized using various techniques. A possible photodegradation mechanism was discussed based on HPLC-MS analysis of degradation products. The study is comprehensive as it covers the process from different aspects and the results are intuitive. A few comments should be considered before consideration for publication as follows:

1.       It seems from SEM that the formed particles constitute smaller primary particles. This can be confirmed by calculating the crystallite size from XRD data using the Scherrer equation as described in this reference 10.3390/catal13101328

2.        Please describe in the text and caption of Fig. 2b that this is the effect of varying the solvothermal temperature.

3.       The histograms in fig. 3 and spectra in Fig 11 are unclear and must be improved to be readable.

4.       The introduction should be strengthened by considering relevant papers on TiO2 for photochemical processes e.g. 10.1016/S1452-3981(23)13982-4, 10.1021/acsomega.2c04576 and authors can refer to this paper (Nanotube Arrays as Photoanodes for Dye Sensitized Solar Cells Using Metal Phthalocyanine Dyes) for supporting the effect of calcination temperature and time on the crystal phase.

5.       The samples names in Fig. 9a should be corrected.

6.       It is reasonable to report rate constant values as only three digits after decimal to reflect the measurement accuracy.

7.       The language of the manuscript should be carefully revised.

Comments on the Quality of English Language

minor editing

Round 2

Reviewer 1 Report

Comments and Suggestions for Authors

The revised version of the manuscript is improved and can be accepted for publication.

Reviewer 2 Report

Comments and Suggestions for Authors

Authors have addressed my concerns and I suggest publication